# FORMAL LIMITATIONS ON THE MEASUREMENT OF MUTUAL INFORMATION

## ABSTRACT

Motivated by applications to unsupervised learning, we consider the problem of measuring mutual information. Recent analysis has shown that naive kNN estimators of mutual information have serious statistical limitations motivating more refined methods. In this paper we prove that serious statistical limitations are inherent to any measurement method. More specifically, we show that any distribution-free high-confidence lower bound on mutual information cannot be larger than $O(\ln N)$ where $N$ is the size of the data sample. We also analyze the Donsker-Varadhan lower bound on KL divergence in particular and show that, when simple statistical considerations are taken into account, this bound can never produce a high-confidence value larger than $\ln N$. While large high-confidence lower bounds are impossible, in practice one can use estimators without formal guarantees. We suggest expressing mutual information as a difference of entropies and using cross entropy as an entropy estimator. We observe that, although cross entropy is only an upper bound on entropy, cross-entropy estimates converge to the true cross entropy at the rate of $1/\sqrt{N}$.

## 1 INTRODUCTION

Motivated by maximal mutual information (MMI) predictive coding (McAllester, 2018; Stratos, 2018; Oord et al., 2018), we consider the problem of measuring mutual information. A classical approach to this problem is based on estimating entropies by computing the average log of the distance to the $k$th nearest neighbor in a sample (Kraskov et al., 2003). It has recently been shown that the classical kNN methods have serious statistical limitations and more refined kNN methods have been proposed (Gao et al., 2014). Here we establish serious statistical limitations on any method of estimating mutual information. More specifically, we show that any distribution-free high-confidence lower bound on mutual information cannot be larger than $O(\ln N)$ where $N$ is the size of the data sample.

Prior to proving the general case, we consider the particular case of the Donsker-Varadhan lower bound on KL divergence (Donsker & Varadhan, 1983; Belghazi et al., 2018). We observe that when simple statistical considerations are taken into account, this bound can never produce a high-confidence value larger than $\ln N$. Similar comments apply to lower bounds based on contrastive estimation. The contrastive estimation lower bound given in Oord et al. (2018) does not establish mutual information of more than $\ln k$ where $k$ is number of negative samples used in the contrastive choice.

The difficulties arise in cases where the mutual information $I(x, y)$ is large. Since $I(x, y) = H(y) - H(y|x)$ we are interested in cases where $H(y)$ is large and $H(y|x)$ is small. For example consider the mutual information between an English sentence and its French translation. Sampling English and French independently will (almost) never yield two sentences where one is a plausible translation of the other. In this case the DV bound is meaningless and contrastive estimation is trivial. In this example we need a language model for estimating $H(y)$ and a translation model for estimating $H(y|x)$. Language models and translation models are both typically trained with cross-entropy loss. Cross-entropy loss can be used as an (upper bound) estimate of entropy and we get an estimate of mutual information as a difference of cross-entropy estimates. Note that the upper-bound guarantee for the cross-entropy estimator yields neither an upper bound nor a lower bound guarantee

for a difference of entropies. Similar observations apply to measuring the mutual information for pairs of nearby frames of video or pairs of sound waves for utterances of the same sentence.

We are motivated by the problem of maximum mutual information predictive coding (McAllester, 2018; Stratos, 2018; Oord et al., 2018). One can formally define a version of MMI predictive coding by considering a population distribution on pairs $(x, y)$ where we think of $x$ as past raw sensory signals (images or sound waves) and $y$ as a future sensory signal. We consider the problem of learning stochastic coding functions $C_x$ and $C_y$ so as to maximize the mutual information $I(C_x(x), C_y(y))$ while limiting the entropies $H(C_x(x))$ and $H(C_y(y))$. The intuition is that we want to learn representations $C_x(x)$ and $C_y(y)$ that preserve "signal" while removing "noise". Here signal is simply defined to be a low entropy representation that preserves mutual information with the future. Forms of MMI predictive coding have been independently introduced in (McAllester, 2018) under the name "information-theoretic cotraining" and in (Oord et al., 2018) under the name "contrastive predictive coding". It is also possible to interpret the local version of DIM (DIM(L)) (Hjelm et al., 2018) as a variant of MMI predictive coding.

A closely related framework is the information bottleneck (Tishby et al., 2000). Here one again assumes a population distribution on pairs $(x, y)$. The objective is to learn a stochastic coding function $C_x$ so as to maximize $I(C_x(x), y)$ while minimizing $I(C_x(x), x)$. Here one does not ask for a coding function on $y$ and one does not limit $H(C_x(x))$.

Another related framework is INFOMAX (Linsker, 1988; Bell & Sejnowski, 1995; Hjelm et al., 2018). Here we consider a population distribution on a single random variable $x$. The objective is to learn a stochastic coding function $C_x$ so as to maximize the mutual information $I(x, C_x(x))$ subject to some constraint or additional objective.

As mentioned above, in cases where $I(C_x(x), C_y(y))$ is large it seems best to train a model of the marginal distribution of $P(C_y)$ and a model of the conditional distribution $P(C_y|C_x)$ where both models are trained with cross-entropy loss. Section 5 gives various high confidence upper bounds on cross-entropy loss for learned models. The main point is that, unlike lower bounds on entropy, high-confidence upper bounds on cross-entropy loss can be guaranteed to be close to the true cross entropy.

Out theoretical analyses will assume discrete distributions. However, there is no loss of generality in this assumption. Rigorous treatments of probability (measure theory) treat integrals (either Riemann or Lebesgue) as limits of increasingly fine binnings. A continuous density can always be viewed as a limit of discrete distributions. Although our proofs are given for discrete case, all our formal limitations on the measurement of mutual information apply to continuous case as well. See Marsh (2013) for a discussion of continuous information theory. Additional comments on this point are given in section 4.

## 2 THE DONSKER-VARADHAN LOWER BOUND

Mutual information can be written as a KL divergence.
$$I(X, Y) = KL(P_{X,Y}, P_X P_Y)$$
Here $P_{X,Y}$ is a joint distribution on the random variables $X$ and $Y$ and $P_X$ and $P_Y$ are the marginal distributions on $X$ and $Y$ respectively. The DV lower bound applies to KL-divergence generally. To derive the DV bound we start with the following observation for any distributions $P$, $Q$, and $G$ on the same support. Our theoretical analyses will assume discrete distributions.

$$
\begin{aligned}
KL(P, Q) &= E_{z \sim P} \ln \frac{P(z)}{Q(z)} \\
&= E_{z \sim P} \ln \left( \frac{G(z)}{Q(z)} \frac{P(z)}{G(z)} \right) \\
&= E_{z \sim P} \ln \frac{G(z)}{Q(z)} + KL(P, G) \\
&\geq E_{z \sim P} \ln \frac{G(z)}{Q(z)} \quad\quad\quad (1)
\end{aligned}
$$

Note that (1) achieves equality for $G(z) = P(z)$ and hence we have

$$KL(P, Q) = \sup_G \ E_{z \in P} \ \ln \frac{G(z)}{Q(z)} \tag{2}$$

Here we can let $G$ be a parameterized model such that $G(z)$ can be computed directly. However, we are interested in $KL(P_{X,Y}, P_X P_Y)$ where our only access to the distribution $P$ is through sampling. If we draw a pair $(x, y)$ and ignore $y$ we get a sample from $P_X$. We can similarly sample from $P_Y$. So we are interested in a KL-divergence $KL(P, Q)$ where our only access to the distributions $P$ and $Q$ is through sampling. Note that we cannot evaluate (1) by sampling from $P$ because we have no way of computing $Q(z)$. But through a change of variables we can convert this to an expression restricted to sampling from $Q$. More specifically we define $G(z)$ in terms of an unconstrained function $F(z)$ as

$$G(z) = \frac{1}{Z} \ Q(z) e^{F(z)} \qquad\qquad Z = \sum_z Q(z) e^{F(z)} = E_{z \sim Q} \ e^{F(z)} \tag{3}$$

Substituting (3) into (2) gives

$$KL(P, Q) = \sup_F \ E_{z \sim P} \ F(z) - \ln E_{z \sim Q} \ e^{F(z)} \tag{4}$$

Equation (4) is the Donsker-Varadhan lower bound. Applying this to mutual information we get

$$\begin{aligned} I(X, Y) \ &= \ KL(P_{X,Y}, P_X P_Y) \\ \\ &= \ \sup_F \ E_{x,y \sim P_{X,Y}} \ F(x, y) - \ln E_{x \sim P_X, \ y \sim P_Y} \ e^{F(x,y)} \end{aligned} \tag{5}$$

This is the equation underlying the MINE approach to maximizing mutual information (Belghazi et al., 2018). It would seem that we can estimate both terms in (5) through sampling and be able to maximize $I(X, Y)$ by stochastic gradient ascent on this lower bound.

## 3 STATISTICAL LIMITATIONS OF KL-DIVERGENCE LOWER BOUNDS

In this section we show that the DV bound (4) cannot be used to measure KL-divergences of more than tens of bits. In fact we will show that no high-confidence distribution-free lower bound on KL divergence can be used for this purpose.

As a first observation note that (4) involves $E_{z \sim Q} \ e^{F(z)}$. This expression has the same form as the moment generating function used in analyzing large deviation probabilities. The utility of expectations of exponentials in large deviation theory is that such expressions can be dominated by extremely rare events (large deviations). The rare events dominating the expectation will never be observed by sampling from $Q$. It should be noted that the optimal value for $F(z)$ in (4) is $\ln(P(z)/Q(z))$ in which case the right hand side of (4) simplifies to $KL(P, Q)$. But for large KL divergence we will have that $F(z) = \ln(P(z)/Q(z))$ is typically hundreds of bits and this is exactly the case where $E_{z \sim Q} \ e^{F(z)}$ cannot be measured by sampling from $Q$. If $E_{z \sim Q} \ e^{F(z)}$ is dominated by events that will never occur in sampling from $Q$ then the optimization of $F$ through the use of (4) and sampling from $Q$ cannot possibly lead to a function $F(z)$ that accurately models the desired function $\ln(P(z)/Q(z))$.

To quantitatively analyze the risk of unseen outlier events we will make use of the following simple lemma where we write $P_{z \sim Q}(\Phi[z])$ for the probability over drawing $z$ from $Q$ that the statement $\Phi[z]$ holds.

**Outlier Risk Lemma:** For a sample $S \sim Q^N$ with $N \geq 2$, and a property $\Phi[z]$ such that $P_{z \sim Q}(\Phi[z]) \leq 1/N$, the probability over the draw of $S$ that no $z \in S$ satisfies $\Phi[z]$ is at least $1/4$.

**Proof:** The probability that $\Phi[z]$ is unseen in the sample is at least $(1 - 1/N)^N$ which is at least $1/4$ for $N \geq 2$ and where we have $\lim_{N \to \infty} (1 - 1/N)^N = 1/e$. Q.E.D.

We can use the outlier risk lemma to perform a quantitative risk analysis of the DV bound (4). We can rewrite (4) as

$$
\begin{aligned}
KL(P, Q) &\geq B(P, Q, F) \\
B(P, Q, F) &= E_{z \sim P} \ F(z) - \ln E_{z \sim Q} \ e^{F(z)}
\end{aligned}
$$

We can try to estimate $B(P, Q, G)$ from samples $S_P$ and $S_Q$, each of size $N$, from the population distributions $P$ and $Q$ respectively.

$$
\hat{B}(S_P, S_Q, F) = \frac{1}{N} \sum_{z \in S_P} F(z) - \ln \frac{1}{N} \sum_{z \in S_Q} e^{F(z)}
$$

While $B(P, Q, F)$ is a lower bound on $KL(P, Q)$, the sample estimate $\hat{B}(S_P, S_Q, F)$ is not. To get a high confidence lower bound on $KL(P, Q)$ we have to handle unseen outlier risk. For a fair comparison with our analysis of cross-entropy estimators in section 5, we will limit the outlier risk by bounding $F(z)$ to the interval $[0, F_{\max}]$. The largest possible value of $\hat{B}(S_P, S_q, F)$ occurs when $F(z) = F_{\max}$ for all $z \in S_P$ and $F(z) = 0$ for all $z \in S_Q$. In this case we get $\hat{B}(S_P, S_Q, F) = F_{\max}$. But by the outlier risk lemma there is still at least a 1/4 probability that

$$
E_{z \sim Q} \ e^{F(z)} \geq \frac{1}{N} e^{F_{\max}}. \tag{6}
$$

Any high confidence lower bound $\tilde{B}(S_P, S_Q, F)$ must account for the unseen outlier risk. In particular we must have

$$
\begin{aligned}
\tilde{B}(S_P, S_Q, F) &\leq F_{\max} - \ln \frac{e^{F_{\max}}}{N} \\
\\
&= \ln N
\end{aligned}
$$

Our negative results can be strengthened by considering the preliminary bound (1) where $G(z)$ is viewed as a model of $P(z)$. We can consider the extreme case of perfect modeling of the population $P$ with a model $G(z)$ where $G(z)$ is computable. In this case we have essentially complete access to the distribution $P$. But even in this setting we have the following negative result.

**Theorem 1** *Let B be any distribution-free high-confidence lower bound on KL(P,Q) computed with complete knowledge of P but only a sample from Q.*

*More specifically, let $B(P, S, \delta)$ be any real-valued function of a distribution $P$, a multiset $S$, and a confidence parameter $\delta$ such that, for any $P$, $Q$ and $\delta$, with probability at least $(1 - \delta)$ over a draw of $S$ from $Q^N$ we have*

$$
KL(P, Q) \geq B(P, S, \delta).
$$

*For any such bound, and for $N \geq 2$, with probability at least $1 - 4\delta$ over the draw of $S$ from $Q^N$ we have*

$$
B(P, S, \delta) \leq \ln N.
$$

**Proof**. Consider distributions $P$ and $Q$ and $N \geq 2$. Define $\tilde{Q}$ by

$$
\tilde{Q}(z) = \left(1 - \frac{1}{N}\right) Q(z) + \frac{1}{N} P(z).
$$

We now have $KL(P, \tilde{Q}) \leq \ln N$. We will prove that from a sample $S \sim Q^N$ we cannot reliably distinguish between $Q$ and $\tilde{Q}$.

We first note that by applying the high-confidence guarantee of the bound to $\tilde{Q}$ have

$$
P_{S \sim \tilde{Q}^N}(B(P, S, \delta) \leq KL(P, \tilde{Q})) \geq 1 - \delta.
$$

The distribution $\tilde{Q}$ equals the marginal on $z$ of a distribution on pairs $(s, z)$ where $s$ is the value of Bernoulli variable with bias $1/N$ such that if $s = 1$ then $z$ is drawn from $P$ and otherwise $z$ is drawn

from $Q$. By the outlier risk lemma the probability that all coins are zero is at least 1/4. Conditioned on all coins being zero the distributions $\tilde{Q}^N$ and $Q^N$ are the same. Let $\mathrm{Pure}(S)$ represent the event that all coins are 0 and let $\mathrm{Small}(S)$ represent the event that $B(P, S, \delta) \leq \ln N$. We now have

$$
\begin{aligned}
P_{S \sim Q^N}(\mathrm{Small(S)}) &= P_{S \sim \tilde{Q}^N}(\mathrm{Small}(S)|\mathrm{Pure}(S)) \\[2mm]
&= \frac{P_{S \sim \tilde{Q}^N}(\mathrm{Pure}(S) \wedge \mathrm{Small(S)})}{P_{S \sim \tilde{Q}^N}(\mathrm{Pure}(S))} \\[2mm]
&\geq \frac{P_{S \sim \tilde{Q}^N}(\mathrm{Pure}(S)) - P_{S \sim \tilde{Q}^N}(\neg\mathrm{Small}(S))}{P_{S \sim \tilde{Q}^N}(\mathrm{Pure}(S))} \\[2mm]
&\geq \frac{P_{S \sim \tilde{Q}^N}(\mathrm{Pure}(S)) - \delta}{P_{S \sim \tilde{Q}^N}(\mathrm{Pure}(S))} \\[2mm]
&= 1 - \frac{\delta}{P_{S \sim \tilde{Q}^N}(\mathrm{Pure}(S))} \\[2mm]
&\geq 1 - 4\delta.
\end{aligned}
$$

## 4 STATISTICAL LIMITATIONS ON ENTROPY LOWER BOUNDS

Mutual information is a special case of KL-divergence. It is possible that tighter lower bounds can be given in this special case. In this section we show similar limitations on lower bounding mutual information. We first note that a lower bound on mutual information implies a lower bound on entropy. The mutual information between $X$ and $Y$ cannot be larger than information content of $X$ alone.

$$I(X, Y) = H(X) - H(X|Y) \leq H(X)$$

So a lower bound on $I(X, Y)$ gives a lower bound on $H(X)$. We show that any distribution-free high-confidence lower bound on entropy requires a sample size exponential in the size of the bound.

The above argument seems problematic for the case of continuous densities as differential entropy can be negative. However, for the continuous case we have

$$I(x, y) = \sup_{C_x, C_y} I(C_x(x), C_y(y))$$

where $C_x$ and $C_y$ range over all maps from the underlying continuous space to discrete sets (all binnings of the continuous space). Hence an $O(\ln N)$ upper bound on the measurement of mutual information for the discrete case applies to the continuous case as well.

The type of a sample $S$, denoted $\mathcal{T}(S)$, is defined to be a function on positive integers (counts) where $\mathcal{T}(S)(i)$ is the number of elements of $S$ that occur $i$ times in $S$. For a sample of $N$ draws we have $N = \sum_i i\mathcal{T}(S)(i)$. The type $\mathcal{T}(S)$ contains all information relevant to estimating the actual probability of the items of a given count and of estimating the entropy of the underlying distribution. The problem of estimating distributions and entropies from sample types has been investigated by various authors (McAllester & Schapire, 2000; Orlitsky et al., 2003; Orlitsky & Suresh, 2015; Arora et al., 2018). Here we give the following negative result on lower bounding the entropy of a distribution by sampling.

**Theorem 2** *Let $B$ be any distribution-free high-confidence lower bound on $H(P)$ computed from a sample type $\mathcal{T}(S)$ with $S \sim P^N$.*

*More specifically, let $B(\mathcal{T}, \delta)$ be any real-valued function of a type $\mathcal{T}$ and a confidence parameter $\delta$ such that for any $P$, with probability at least $(1 - \delta)$ over a draw of $S$ from $P^N$, we have*

$$H(P) \geq B(\mathcal{T}(S), \delta).$$

*For any such bound, and for $N \geq 50$ and $k \geq 2$, with probability at least $1 - \delta - 1.01/k$ over the draw of $S$ from $P^N$ we have*

$$B(\mathcal{T}(S), \delta) \leq \ln 2kN^2.$$

**Proof:** Consider a distribution $P$ and $N \geq 100$. If the support of $P$ has fewer than $2kN^2$ elements then $H(P) < \ln 2kN^2$ and by the premise of the theorem we have that, with probability at least $1 - \delta$ over the draw of $S$, $B(\mathcal{T}(S), \delta) \leq H(P)$ and the theorem follows. If the support of $P$ has at least $2kN^2$ elements then we sort the support of $P$ into a (possibly infinite) sequence $x_1, x_2, x_3, \ldots$ so that $P(x_i) \geq P(x_{i+1})$. We then define a distribution $\tilde{P}$ on the elements $x_1, \ldots, x_{2kN^2}$ by

$$\tilde{P}(x_i) = \begin{pmatrix} P(x_i) & \text{for } i \leq kN^2 \\ \\ \frac{P(i > kN^2)}{kN^2} & \text{for } kN^2 < i \leq 2kN^2 \end{pmatrix}$$

We will let $\mathrm{Small}(S)$ denote the event that $B(\mathcal{T}(S), \delta) \leq \ln 2kN^2$ and let $\mathrm{Pure}(S)$ abbreviate the event that no element $x_i$ for $i > kN^2$ occurs twice in the sample. Since $\tilde{P}$ has a support of size $2kN^2$ we have $H(\tilde{P}) \leq \ln 2kN^2$. Applying the premise of the lemma to $\tilde{P}$ gives

$$P_{S \sim \tilde{P}^N}(\mathrm{Small}(S)) \geq 1 - \delta \tag{7}$$

For a type $\mathcal{T}$ let $P_{S \sim P^N}(\mathcal{T})$ denote the probability over drawing $S \sim P^N$ that $\mathcal{T}(S) = \mathcal{T}$. We now have

$$P_{S \sim P^N}(\mathcal{T}|\mathrm{Pure}(S)) = P_{S \sim \tilde{P}^N}(\mathcal{T}|\mathrm{Pure}(S)).$$

This gives the following.

$$\begin{aligned} P_{S \sim P^N}(\mathrm{Small}(S)) &\geq P_{S \sim P^N}(\mathrm{Pure}(S) \wedge \mathrm{Small}(S)) \\ &= P_{S \sim P^N}(\mathrm{Pure}(S)) \, P_{S \sim P^N}(\mathrm{Small}(S) \mid \mathrm{Pure}(S)) \\ &= P_{S \sim P^N}(\mathrm{Pure}(S)) \, P_{S \sim \tilde{P}^N}(\mathrm{Small}(S) \mid \mathrm{Pure}(S)) \\ &\geq P_{S \sim P^N}(\mathrm{Pure}(S)) \, P_{S \sim \tilde{P}^N}(\mathrm{Pure}(S) \wedge \mathrm{Small}(S)) \end{aligned} \tag{8}$$

For $i > kN^2$ we have $\tilde{P}(x_i) \leq 1/(kN^2)$ which gives

$$P_{S \sim \tilde{P}^N}(\mathrm{Pure}(S)) \geq \prod_{j=1}^{N-1} \left(1 - \frac{j}{kN^2}\right)$$

Using $(1 - P) \geq e^{-1.01 \, P}$ for $P \leq 1/100$ we have the following birthday paradox calculation.

$$\begin{aligned} \ln P_{S \sim \tilde{P}^N}(\mathrm{Pure}(S)) &\geq -\frac{1.01}{kN^2} \sum_{j=1}^{N-1} j \\ &= -\frac{1.01}{kN^2} \frac{(N-1)N}{2} \\ &\geq -.505/k \\ P_{S \sim \tilde{P}^N}(\mathrm{Pure}(S)) &\geq e^{-.505/k} \geq 1 - .505/k \end{aligned} \tag{9}$$

Applying the union bound to (7) and (9) gives.

$$P_{S \sim \tilde{P}^N}(\mathrm{Pure}(S) \wedge \mathrm{Small}(S)) \geq 1 - \delta - .505/k \tag{10}$$

By a derivation similar to that of (9) we get

$$P_{S \sim P^N}(\mathrm{Pure}(S)) \geq 1 - .505/k \tag{11}$$

Combining (8), (10) and (11) gives

$$P_{S \sim P^N}(\mathrm{Small}(S)) \geq 1 - \delta - 1.01/k$$

## 5 Cross Entropy as an Entropy Estimator

Since mutual information can be expressed as a difference of entropies, the problem of measuring mutual information can be reduced to the problem of measuring entropies. In this section we show that, unlike high-confidence distribution-free lower bounds, high-confidence distribution-free upper bounds on entropy can approach the true cross entropy at modest sample sizes even when the true cross entropy is large. More specifically we consider the cross-entropy upper bound.

$$
\begin{aligned}
H(P) &= E_{x \sim P} \ln \frac{1}{P(x)} \\
&= E_{x \sim P} \ln \left( \frac{1}{G(x)} \frac{G(x)}{P(x)} \right) \\
&= H(P, G) - KL(P, G) \\
&\leq H(P, G)
\end{aligned}
$$

For $G = P$ we get $H(P, G) = H(P)$ and hence we have

$$
H(P) = \inf_G H(P, G)
$$

In practice $P$ is a population distribution and $G$ is model of $P$. For example $P$ might be a population distribution on paragraphs and $G$ might be an autoregressive RNN language model. In practice $G$ will be given by a network with parameters $\Phi$. In this setting we have the following upper bound entropy estimator.

$$
\hat{H}(P) = \inf_\Phi H(P, G_\Phi) \tag{12}
$$

The gap between $\hat{H}(P)$ and $H(P)$ depends on the expressive power of the model class.

The statistical limitations on distribution-free high-confidence lower bounds on entropy do not arise for cross-entropy upper bounds. For upper bounds we can show that naive sample estimates of the cross-entropy loss produce meaningful (large entropy) results. We first define the cross-entropy estimator from a sample $S$.

$$
\hat{H}(S, G) = \frac{1}{|S|} \sum_{x \in S} -\ln G(x)
$$

We can bound the loss of a model $G$ by ensuring a minimum probability $e^{-F_{\max}}$ where $F_{\max}$ is then the maximum possible log loss in the cross-entropy objective. In language modeling a loss bound exists for any model that ultimately backs off to a uniform distribution on characters. Given a loss bound of $F_{\max}$ we have that $\hat{H}(S, G)$ is just the standard sample mean estimator of an expectation of a bounded variable. In this case we have the following standard confidence interval.

**Theorem 3** *For any population distribution $P$, and model distribution $G$ with $-\ln G(x)$ bounded to the interval $[0, F_{\max}]$, with probability at least $1 - \delta$ over the draw of $S \sim P^N$ we have*

$$
H(P, G) \in \hat{H}(S, G) \pm F_{\max} \sqrt{\frac{\ln \frac{2}{\delta}}{2N}}
$$

It is also possible to give PAC-Bayesian bounds on $H(P, G_\Phi)$ that take into account the fact that $G_\Phi$ is typically trained so as to minimize the empirical loss on the training data. The PAC-Bayesian bounds apply to "broad basin" losses and loss estimates such as the following.

$$
\begin{aligned}
H_\sigma(S, G_\Phi) &= E_{x \sim P} \, E_{\epsilon \sim N(0, \sigma I)} \, -\ln G_{\Phi + \epsilon}(x) \\
\hat{H}_\sigma(S, G_\Phi) &= \frac{1}{|S|} \sum_{x \in S} E_{\epsilon \sim N(0, \sigma I)} \, -\ln G_{\Phi + \epsilon}(x)
\end{aligned}
$$

Under mild smoothness conditions on $G_\Phi(x)$ as a function of $\Phi$ we have

$$\lim_{\sigma \to 0} H_\sigma(P, G_\Phi) = H(P, G_\Phi)$$

$$\lim_{\sigma \to 0} \hat{H}_\sigma(S, G_\Phi) = \hat{H}(S, G_\Phi)$$

An L2 PAC-Bayesian generalization bound (McAllester (2013)) gives that for any parameterized class of models and any bounded notion of loss, and any $\lambda > 1/2$ and $\sigma > 0$, with probability at least $1 - \delta$ over the draw of $S$ from $P^N$ we have the following simultaneously for all parameter vectors $\Phi$.

$$H_\sigma(P, G_\Phi) \leq \frac{1}{1 - \frac{1}{2\lambda}} \left( \hat{H}_\sigma(S, G_\Phi) + \frac{\lambda F_{\max}}{N} \left( \frac{||\Phi||^2}{2\sigma^2} + \ln \frac{1}{\delta} \right) \right)$$

It is instructive to set $\lambda = 5$ in which case the bound becomes.

$$H_\sigma(P, G_\Phi) \leq \frac{10}{9} \left( \hat{H}_\sigma(S, G_\Phi) + \frac{5 F_{\max}}{N} \left( \frac{||\Phi||^2}{2\sigma^2} + \ln \frac{1}{\delta} \right) \right)$$

While this bound is linear in $1/N$, and tighter in practice than square root bounds, note that there is a small residual gap when holding $\lambda$ fixed at 5 while taking $N \to \infty$. In practice the regularization parameter $\lambda$ can be tuned on holdout data. One point worth noting is the form of the dependence of the regularization coefficient on $F_{\max}$, $N$ and the basin parameter $\sigma$.

It is also worth noting that the bound can be given in terms of "distance traveled" in parameter space from an initial (random) parameter setting $\Phi_0$.

$$H_\sigma(P, G_\Phi) \leq \frac{10}{9} \left( \hat{H}_\sigma(S, G_\Phi) + \frac{5 F_{\max}}{N} \left( \frac{||\Phi - \Phi_0||^2}{2\sigma^2} + \ln \frac{1}{\delta} \right) \right)$$

Evidence is presented in Dziugaite & Roy (2017) that the distance traveled bounds are tighter in practice than traditional L2 generalization bounds.

## 6 MMI PREDICTIVE CODING

Recall that in MMI predictive coding we assume a population distribution on pairs $(x, y)$ where we think of $x$ as past raw sensory signals (images or sound waves) and $y$ as a future sensory signal. We then consider the problem of learning stochastic coding functions $C_x$ and $C_y$ that maximizes the mutual information $I(C_x(x), C_y(y))$ while limiting the entropies $H(C_x(x))$ and $H(C_y(y))$. Here we propose representing the mutual information as a difference of entropies.

$$I(C_x(x), C_y(y)) = H(C_y(y)) - H(C_y(y)|C_x(x))$$

When the coding functions are parameterized by a function $\Psi$, the above quantities become a function of $\Psi$. We can then formulate the following nested optimization problem.

$$\Psi^* = \operatorname*{argmax}_\Psi \hat{H}(C_y(y); \Psi) - \hat{H}(C_y(y)|C_x(x); \Psi)$$

$$\hat{H}(C_y(y); \Psi) = \inf_\Theta H(C_y(y), G_\Theta; \Psi)$$

$$\hat{H}(C_y(y)|C_x(x); \Psi) = \inf_\Phi H(C_y(y), G_\Phi|C_x(x); \Psi)$$

The above quantities are expectations over the population distribution on pairs $(x, y)$. In practice we have only a finite sample form the population. But the preceding section presents theoretical evidence that, unlike lower bound estimators, upper bound cross-entropy estimators can meaningfully estimate large entropies from feasible samples.

## 7 CONCLUSIONS

Maximum mutual information (MMI) predictive coding seems well motivated as a method of unsupervised pretraining of representations that maintain semantic signal while dropping uninformative noise. However, the maximization of mutual information is a difficult training objective. We have given theoretical arguments that representing mutual information as a difference of entropies, and estimating those entropies by minimizing cross-entropy loss, is a more statistically justified approach than maximizing a lower bound on mutual information.

Unfortunately cross-entropy upper bounds on entropy fail to provide either upper or lower bounds on mutual information — mutual information is a difference of entropies. We cannot rule out the possible existence of superintelligent models, models beyond current expressive power, that dramatically reduce cross-entropy loss. Lower bounds on entropy can be viewed as proofs of the non-existence of superintelligence. We should not surprised that such proofs are infeasible.

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
