# OpenReview forum: "Formal Limitations on the Measurement of Mutual Information"
_ICLR.cc/2019/Conference_

### Official Review · AnonReviewer3 · 2018-10-16
**ICLR 2019 Conference Paper277 AnonReviewer3**

**Rating:** 4
**Confidence:** 4

**Review:**

This paper studied the Donsker-Varadhan lower bound of KL-divergence. The authors show that with high probability, the DV lower bound is upper bounded by log of the sample size, so if the true KL-divergence is very large, then exponential sample size is needed to make the DV lower bound tight. The same argument holds true for any distribution-free high-confidence lower bound (such as DV lower bound) for KL divergence. Then the authors proposed to use an upper bound for entropy instead of lower bound for mutual information.

The idea of the paper is interesting and the proof of Theorems 1 and 2 are valid. Especially I like the idea of Theorem 1, which proves that any distribution-free high-confidence lower bound for KL divergence is upper bounded by log of sample size. This idea is similar to the paper in (Gao et al 15') which shows that mutual information estimator is upper bounded by log(N).

However, this paper contains many fatal flaws, which significantly weaken the quality of this paper. Precisely,

1. The DV lower bound is just an alternative of mutual information, helping MMI predictive coding algorithm to find good coding functions C_x and C_y. The goal of MMI predictive coding is not to estimate the mutual information I(C_x(x), C_y(y)) precisely, instead, the goal is to find good coding functions. The fact that DV lower bound is small means that we can not estimate mutual information through DV lower bound, but it does not directly imply that we can not find the coding functions. I expect some experiments to show that when mutual information is large, MMI predictive coding using DV lower bound can not find good coding functions.

2. In Section 3, the formula after the proof of Outlier Risk Lemma and before Theorem 1 (btw, it is better to have numbers for these formula) seems to be problematic. The first formula shows that E_{z~q} e^{F(z)} >= (1/N)e^{F_max}, then we plug it in (4). But in (4) there is negative ln of E_{z~q} e^{F(z)}, so we should have KL(P,Q) <= something, correct? This may be a typo but this typo is so important such that it affect the readability a lot. Theorem 1 is correct but the paragraphs before Theorem 1 confuse the reader a lot.

3. In Section 4, are you considering classical entropy for discrete random variables, or differential entropy for continuous random variables? I assume you are considering the latter, since most people of machine learning community are interested in continuous random variables. Then your statement of I(X;Y) <= H(X) is incorrect, since for continuous random variables, H(X|Y) can be negative. See (Thomas & Cover, Chapter 8) for a reference.

4. Related to problem 3, if you are considering continuous random variables, then the statement I(X;Y)=H(X)+H(Y)-H(X,Y) is not always correct. There are cases that H(X) is infinite, H(Y) is infinite, H(X,Y) is infinite but I(X;Y) is finite. These cases does not only exist in mathematical books, but also exists in practice, especially when the data is located on a low-dimensional manifold embedded in a high-dimensional space. Therefore, your approach of decompose the mutual information is not always possible.

5. Regarding to your proposed optimization problem in Section 6 (also it is better to have a number), I have some concerns. Since it involves max over \Psi outside, and inf over \Theta and inf over \Phi inside, so I wonder how do you solve this problem? Can you guarantee that the solution can provide good coding functions C_x and C_y? Also, it seems that this optimization problem is proposed as an improvement over the DV lower bound method, so I wish to see some experiment showing that this method is better than the DV lower bound method, at least for some synthetic datasets.

Because of the above mentioned flaws (especially 3 and 4, and lack of experiments), I think the paper is below the standard of ICLR conference.

References:
[1] Efficient Estimation of Mutual Information for Strongly Dependent Variables, by Gao, Ver Steeg and Galstyan, AISTATS15'
[2] Elements of Information Theory, 2nd edition, by Thomas and Cover.

---

> ### Comment · Area_Chair1 · 2018-10-16
> **Authors are focusing on discrete distributions for analysis**
>
> On points (3) and (4), I think this is addressed by the statement in Section 2 just before Eq. (1): "Our theoretical analyses will assume discrete distributions."

---

> > ### Comment · AnonReviewer3 · 2018-10-16
> > **Some experiments are needed to justify your theory**
> >
> > If the theory works only for discrete distributions, it is not so interesting to machine learning community, unless you can show that the insights from the analysis for discrete case can be brought to continuous case. So some experiments which show that the method proposed in Section 6 has better performance are needed.

---

> ### Comment · AnonReviewer1 · 2018-11-06
> **discrete v. continuous**
>
> "most people of machine learning community are interested in continuous random variables"
> - This may be technically true, but a very many machine learning problems involve categorical data, i.e. discrete valued data. Considering only discrete-valued data would never make it irrelevant to the "machine learning community" any more than considering only continuous-valued data would.

---

> ### Author Response · Authors · 2018-11-12
> **Most Points Addressed**
>
> 1. I believe that it is important to show that theory can have predictive power.  It can be useful to publish theory prior to its verification.  The theorems seems nontrivial and important on their own.  The machine translation example also seems compelling without experiments.
>
> 2. The original formulas were correct but unclear.  This section has been rewritten.
>
> 3 & 4.  As noted above, the results apply to the continuous case.  The particular issue with negative differential entropy is addressed by noting the following equality
> I(x,y) = sup_{C_x,C_y}  I(C_x(x),C_y(y))
> where C_x and C__y range over all discrete classifications (binnings) of the continuous space.  This follows
> from either the Reimann or Lebesgue definition of the integral defining expectation together with the data-processing inequality for mutual information in the discrete case.  A sophisticated discussion of continuous information theory can be found at
> http://www.crmarsh.com/static/pdf/Charles_Marsh_Continuous_Entropy.pdf
>
> 5.  This a legitimate issue.  The optimization problem is adversarial as in GANs.  This indeed raises concerns about the stability of the optimization.  Note, however, that the adversarial objective only involves the marginal entropy H(C_y(y)) --- the parameters governing H(C_y(y)|C_x(x)) are cooperative.   In the translation example the adversarial part is the model of the background distribution of sentence codes.  My intuition is that the background model should be more stable than the conditional model yielding stable adversarial training.  Also, GANs do seem to be able to manage adversarial optimization.  The revision does not discuss this issue.

---

### Official Review · AnonReviewer2 · 2018-11-03
**Promising work from theoretical standpoint**

**Rating:** 6
**Confidence:** 3

**Review:**

This paper is about estimating mutual information in high dimensional settings. This is a very challenging open problem, that is of interest to a diverse set of research communities.

In this paper, it is theoretically argued that the recent proposed mutual information (lower bound) estimator, MINE, that is based on the Donsker-Varadhan representation of the corresponding KL divergence expression for mutual infortmation, is fundamentally flawed for high dimensions (of discrete variables).  It is further shown that lower bounds for joint entropy are hard to obtain due to exponential sample complexity. So, the authors suggest to instead obtain an upper bound for each entropy term in the mutual information expression; cross-entropy is the suggested upper bound for an entropy term.

I have some basic questions as the following.

Since the recent KL divergence based MI estimator, MINE, is inaccurate in high dimensions, there should be at least a discussion on connections between your estimator and the classic nearest neighbor distances based estimator of Kraskov et al. and the extensions (I suppose, even for discrete variables, one can compute distances to obtain nearest neighbors efficiently). Also, there are kernel functions based estimators.

There is no discussion in the paper about the errors accumulating from individual entropy terms in the mutual information expression. Kraskov et al. talk about this problem of accumulating errors in their seminal paper and propose not to compute the entropy terms individually. What you are proposing is in contrast to their clever observations.

Does the analysis on upper bound for entropy term also apply to the conditional entropy in the mutual information expression ? I think, there are more subtleties that should be explained.

Since the proposed approach upper bounds entropy using cross entropy term (i.e. using some machine learning model like a neural network), it is even more important to show solid empirical evaluation, for synthetic as well as real world data.

There is a subtle difference between estimating mutual information and proposing an upper/lower bound for it. At present, it is not clear if the proposed upper bound of entropy would lead to an overall upper bound or lower bound for the mutual infortmation expression. The latter is important to know both in the context of optimization based on mutual information maximization (it should be lower bound in such case), as well analyzing mutual infortmation to under complex dynamics such as in brain.

---

> ### Comment · AnonReviewer1 · 2018-11-06
> **upper/lower bounds**
>
> "At present, it is not clear if the proposed upper bound of entropy would lead to an overall upper bound or lower bound for the mutual infortmation expression."
>
> -The authors addressed this in their conclusion:
> Unfortunately cross entropy upper bounds on entropy fail to provide either upper or lower bounds on
> mutual information—mutual information is a difference of entropies. We cannot rule out the possible
> existence of superintelligent models, models beyond current expressive power, that dramatically
> reduce cross-entropy loss. Lower bounds on entropy can be viewed as proofs of the non-existence
> of superintelligence. We should not surprised that such proofs are infeasible.

---

> ### Author Response · Authors · 2018-11-12
> **Most Points Addressed**
>
> The revision now briefly discusses kNN methods but largely just to point out that our results apply to them also, and to the continuous case generally.
>
> We did not explicitly discuss the idea that a difference of two quantities (a difference of entropies) accumulates more error than a direct single-quantity measurement.  However, there is a new example discussed --- the mutual information between an English sentence and its French translation --- in which the difference of entropy approach has very clear advantages.
>
> Yes, the conditional entropy can also be upper bounded by cross-entropy.  We did not discuss this explicitly in the revision but the generalization of a theorem about distributions to a theorem about conditional distributions is generally straightforward.
>
> I think the example of NLP translation makes the empirical point sufficiently clear as cross-entropy is the main method used in practice for training both language models and translation systems.
>
> The revision makes it clear that we do not get either an upper or lower bound on mutual information from the cross-entropy upper bounds on entropy as this is only an upper bound and mutual information is a difference of entropies.

---

> > ### Comment · AnonReviewer2 · 2018-11-22
> > **More analysis required relating to other classical works**
> >
> > In the updated draft, the authors argue that the existing kNN based estimators have serious statistical limitations citing the work of Gao et al. This statement is not correct. The work of Gao et al., has not proved that kNN based estimators are statistically limited. As per theorem 2 and 3 in that paper, the required number of samples to estimate mutual information is proportional to exp(I(.)/d-1) with d denoting data dimension. In typical settings, the ratio of mutual information I(.) and data dimension is a constant. In such settings, the required number of samples is not large. It is only in extreme conditions of very high mutual information, one would require an exponential number of samples. So, for general settings, the classical estimators like kNN, KSG remain relevant.
> >
> > Authors prove that mutual information lower bound can not be more than log N. The question is what is its relation to actual mutual information quantity. In general machine learning settings, mutual information between high dimensional variables is low. In such case, mutual information lower bounds of values log N may be enough, and so it applies to MINE, and classical estimators like kNN, KSG.
> >
> > Also, since their emphasis is on the criticism of MINE that is an estimator for continuous variables, it would serve better to tailor their analysis for those settings. At present, their analysis is for discrete variables. In general, I appreciate works even on mutual information of discrete variables, regardless the value for particular models in ML. Along those lines, it is also worth noting that for discrete variables, it has been proven that sample complexity is S/log(S) where S is support size of the distribution.
> >
> > Regardless, I find this paper very interesting, and promising. However, the present exposition of the work can be misleading to the community. For an instance, the title is discouraging stating as if the present-works on mutual information are irrelevant. This is not true even in the light of this work. Also, for a venue like ICLR, it would be important to provide experimental analysis even if for synthetic data. Overall my impression is that it can be a good work, but preliminary at present.

---

> > > ### Comment · AnonReviewer1 · 2018-11-22
> > > **MI lower bound no more than log N -> implies exponential sample complexity by definition.**
> > >
> > > "The question is what is its relation to actual mutual information quantity. In general machine learning settings, mutual information between high dimensional variables is low."
> > >
> > > Note this is irrelevant to the theory; regardless of the value of I(;), the lower bound implies that you need log N to be at least as large as I(;). For this to occur you need at least N = e^{I(;)} by definition, establishing exponential sample complexity regardless of I(;) being "large" or "small".
> > >
> > > In practice this can be relevant of course, if the true MI is small then exponential sample complexity scaling with I(;) may be ok. I don't see how this detracts from the message of the paper however. i.e. "fundamental limits".

---

### Official Review · AnonReviewer1 · 2018-11-07
**Timely discussion of approaches for approximating mutual information**

**Rating:** 8
**Confidence:** 5

**Review:**

This paper considers the problem of estimating bounds on the mutual information. It begins by showing that popular recent estimators (e.g. MINE) are flawed, since they rely on the Donsker-Varadhan bound that cannot be estimated efficiently. They then point to entropy upper bounds as a much more feasible approach to MI approximation, and propose a framework for using them in practice. These upper bounds converge as 1/sqrt(N) to the true entropy value, making them potentially viable in practice to obtain reasonable approximations of MI.

Given the significant recent attention payed to the MINE estimator and to MI estimation in machine learning generally, I think the message of this paper is very critical to the machine learning community. This analysis of the MINE estimator alone would warrant publication.

There are currently some weaknesses to this paper, however, when compared to the typical ICLR paper. If the following issues are addressed I am prepared to raise my scoring of this paper:
1. Provide some intuition on how to apply this type of analysis to the continuous-valued case, or some reason why such an analysis would require a different framework. The more detail the better, if bounds analogous to Theorem 1 could be proved for continuous variables and added to the paper, that would be excellent.
2. Section 4 as written is intuitively clear, but could greatly benefit from a full rigorous analysis. There is plenty of space in the paper to do this (and the supplement is available if needed). If such an analysis can’t be done, this section should be deleted.
3. Some empirical example of MINE converging slowly in practice would greatly add to the impact of the paper.

Minor issues:
Last sentence of Section 1 should clarify what “true entropy” means. As written it is ambiguous between the actual entropy or the cross-entropy, since both are entropies.

I understand the nested optimization problem in Section 6, but the presentation is somewhat unclearly written. More exposition here would help, along with a more clear step-by-step explanation of the practical procedure.

EDIT: The authors have addressed my concerns and I have raised my score.

---

> ### Author Response · Authors · 2018-11-12
> **Most points addressed.**
>
> The revision has addressed points 1 and 2 but not 3.  An empirical evaluation is not going to happen.  It would be fun to get a pure theory paper into ICLR.

---

> > ### Comment · AnonReviewer1 · 2018-11-12
> > **title**
> >
> > Thanks. This looks sufficient to me, I will raise my score.

---

### Author Response · Authors · 2018-11-10
**Revision Submitted**

We just submitted a revision taking reviewer comments into account.  As explicitly noted in the revision, the results apply to continuous as well as discrete distributions.  Most other reviewer comments are addressed as well.  When I get a chance I will respond to the reviews on a more point-by-point basis.

---

### Meta-Review · Area_Chair1 · 2018-12-13
**Strong theoretical analysis that highlights a weakness in MINE, but some experiments would have been nice**

**Confidence:** 3
**Recommendation:** Reject

**Metareview:**

The paper proves that the Donsker-Varadhan lower bound on KL divergence cannot be used to estimate KL divergences of more than tens of bits, and that more generally any distribution-free high-confidence lower bound on mutual information cannot be larger than O(ln N) where N is the size of the data sample. As an alternative for applications such as maximum mutual information predictive coding, a form of representation learning, the paper proposes using the cross-entropy upper bound on entropy and estimating mutual information as a difference of two cross-entropies. These cross-entropy bounds converge to the true entropy as 1/\sqrt(N), but at the cost of providing neither an upper nor a lower bound on mutual information. There was a divergence of opinion between the reviewers on this paper. The most negative reviewer (R3) thought there should be experiments confirming that the DV bound fails when mutual information is high, was concerned that the theory applied only in the case of discrete distributions, and was concerned that the proposed optimization problem in Section 6 would be challenging due to its adversarial (max-inf) structure. The authors responded that they felt the theory could stand on its own without empirical tests (a point with which R1 agreed); that although their exposition was for discrete variables, the analysis applies to the continuous case as well; and that they agreed with the point about the difficulty of the optimization, but that GANs face similar difficulties. Because R3 did not participate in the discussion and the AC believes that the authors adequately addressed most of R3's issues in their response and revision, this review has been discounted. The next most negative reviewer (R2) wanted a discussion relating the ideas in this paper to kNN and kernel-based estimators of mutual information, wanted an empirical evaluation (like R3), and was concerned about whether the difference of cross-entropies provides an upper or lower bound on mutual information. In their response and revision the authors added some discussion of kNN methods (but not enough to make R2 happy) and clarified that the difference of cross-entropies provides neither an upper nor a lower bound. The most positive reviewer (R1) thinks the theoretical contribution of the paper is significant enough to justify publication in ICLR. The AC likes the theoretical work and feels that it raises important concerns about MINE, but concurs with R2 and R3 that some empirical validation of the theory is needed for the paper to appear in ICLR. The authors are strongly encouraged to perform an empirical validation of the theory and to submit this work to another machine learning venue.